# A Four-Year Follow-Up Case Report of Hypomineralized Primary Second Molars Rehabilitated with Stainless Steel Crowns

**DOI:** 10.3390/children8100923

**Published:** 2021-10-16

**Authors:** Luísa Bandeira Lopes, Vanessa Machado, João Botelho

**Affiliations:** 1Dental Pediatrics Department, Egas Moniz Dental Clinic, Egas Moniz—Cooperativa de Ensino Superior, CRL, 2829-411 Almada, Portugal; 2Clinical Research Unit, Centro de Investigação Interdisciplinar Egas Moniz (CiiEM), Egas Moniz—Cooperativa de Ensino Superior, CRL, 2829-411 Almada, Portugal; vmachado@egasmoniz.edu.pt (V.M.); jbotelho@egasmoniz.edu.pt (J.B.); 3Evidence-Based Hub, Clinical Research Unit, CiiEM, Egas Moniz—Cooperativa de Ensino Superior, CRL, 2829-411 Almada, Portugal

**Keywords:** hypomineralized second primary molars, molar incisor hypomineralization, occlusion, molar relationship, tooth loss, space maintenance, mesial tooth movement

## Abstract

Hypomineralized primary second molars (HPSM) are characterized by enamel opacities accompanied by hypersensitivity and atypical caries lesion, on one to four primary second molars. The correct treatment and follow-ups of those teeth have an important impact on a correct eruption of the first permanent molars and future occlusion. Hence, this report aims to describes a case of a severe HPSM in all second molars of a four-year-old girl and subsequent four-year follow-ups. The rehabilitation involved the placement of four stainless steel crowns on all four second primary molars under general anesthesia. Concerning the available literature and the case severity of HPSM, the treatment approach proposed for the case provided good functional outcome.

## 1. Introduction

Dental development defects (DDDs) are frequently observed in primary and permanent dentitions, although dental caries is the most frequent oral health condition in children worldwide [1]. The group of DDDs can be divided into hypomineralization and hypoplasia. While enamel hypoplasia refers to a quantitative defect of enamel, hypomineralization relies on qualitative deficits [2,3].

The term “Molar-Incisor Hypomineralization” (MIH) was presented in 2003, in Athens, being defined as a systemic hypomineralization of one or more permanent first molars with possible involvement of the incisors. In MIH, teeth are characterized by demarcated creamy-white, yellow or brown opacities, post-eruptive enamel breakdown, hypersensitivity, and atypical caries lesion formation [3,4,5,6]. Later, the European Academy of Paediatric Dentistry (EAPD) officially established this condition as MIH [7]. After these diagnostic criteria, an interim seminar and workshop on MIH was organized in Helsinki in 2009 (Lydidakis, 2010). As a result, a consensus paper on the prevalence, diagnosis, etiology and treatment was proposed. Regarding the diagnostic criteria and clinical appearance of the defects, it was agreed that a case of MIH was diagnosed if at least one permanent first molar had hypomineralization of the enamel. Besides, permanent incisors can also be affected, as well as primary second molars and the tip of canines (Lydidakis, 2010).

Therefore, and since hypomineralization was also reported in the primary second molars, it was defined as hypomineralized primary second molars (HSPM) [5,6,8]. In this sense, the EAPD adopted these diagnostic criteria [1,9,10]. Ever since, multiple studies have studied the etiology of HSPM, pointing to multifactorial, although it is still unclear [2,4,6,8,9,11,12,13,14,15,16,17,18]. In regard to prevalence, HPSM is significantly reported as low, ranging from 2% to 21.8% [1,5,8,10,11,12,14,15,16,18,19], though a recent study estimated to be around 4.9% to 9.0% of the worldwide population [6].

The presence of HPSM has been described as an important risk factor for caries and early loss of primary secondary molars, because the hypomineralized enamel is porous and brittle, causes an enlargement sensitivity to thermal and mechanical stimuli, and causes discomfort during teeth brushing [1,5,9,14,18,20].

The possible treatments modalities for teeth with hypomineralization are based on its severity, with the prevention, restoration, or rehabilitation with stainless steel crowns or even tooth extraction being possible. The decision relies on several factors, like the severity of the case, the patient’s dental age and collaboration. Notwithstanding, the presence of the second primary molars is vital to ensure a future correct occlusal development.

Hence, we present a 4-year follow-up of a severe HSPM case report, which shows and highlights the importance of a full coverage with preformed stainless-steel crowns, and its impact on correct occlusal development.

## 2. Case Presentation

A four-year-old girl attended the Pediatric Dentistry Department at Egas Moniz Dental Clinic (Almada, Portugal). Informed consent was obtained from her parents so that case records could be made available for teaching purposes, including scientific publication. All procedures were carried out in accordance with the Helsinki declaration, as revised in 2013.

The patient presented no relevant medical history, and her mother reported a high thermal sensitivity to cold on the posterior teeth, with chewing difficulty, as well as tooth brushing. During clinical evaluation, an extraoral examination showed no facial asymmetry or swelling. Intraoral and radiographic examinations showed good oral hygiene and extensive enamel breakdown with irregular opacities on all primary second molars, being the differential diagnosis compatible with HPSM (Figure 1). The patient’s parents were instructed to use a GC Tooth Mousse and a toothpaste with 1500 ppm fluoride as a routine at-home oral hygiene practice, until the specific treatment appointment.

Considering her young age, her non-cooperative behavior, the fact that the severity of the HPSM, which includes teeth 65 and 75, had big proximity to the pulp (Figure 2), and the importance of the second primary molars, it was advised to place four crowns on all second primary molars. Two treatment options were presented to the patient’s parents. The first was four zirconia crowns on the four primary molars. However, this was rejected duo to the high economic value, despite the aesthetics, the mechanical resistance, limited plaque adhesion, wear behavior and natural appearance. The second treatment option was four stainless steel crowns. This last option allows the eruption of the first definitive molars, maintains the vertical dimension, and grants the physiological exfoliation of the primary second molars at a lower cost. Firstly, nitrous oxide sedation was proposed, but it was not successful, given the non-cooperative behavior. Therefore, the complete treatment was performed under general anesthesia. After plaque removal with a prophylactic paste at the beginning of the procedure and the gently drying of the tooth, a rubber dam was applied, one tooth at a time, to locally isolate the lesions. Then, and after partial carious lesion removal, stainless steel crowns (3M™ ESPE™ Stainless Steel Crowns, 3 M, St. Paul, MN, USA) were selected and cemented with Ketac™ Cem Easy Mix (3M™ ESPE™, Maplewood, MN, USA), according to the manufacturer’s instructions (Figure 3). Excess cement was removed. The position of the margins was sub-gingival, and both sides of each tooth had contact with adjacent teeth to allow proper oral hygiene.

After a three-month period, we performed a panoramic radiograph (Figure 4), and every six months, clinical and radiograph exams were collected. At the end of four years of follow-up, clinical evaluation revealed a good marginal adaptation and gingival health (Figure 5), and radiographic exam showed normal occlusion in all the first permanent molars (Figure 6). Moreover, at each appointment, information was given regarding gingival health, occlusal contacts, adaptation of crown margins, and the presence of clinical signs of infection. No additional treatment was needed during the follow-up period.

## 3. Discussion

In the present case, we have considered stainless steel crowns in all second primary molars for restorative therapy of a severe case of HPSM. Even though HPSM is a clinical challenge, the maintenance of second primary molars is of the upmost importance to avoid functional and aesthetic negative outcomes. The described case showed an effective treatment until a 4-year follow-up and allowed the correct eruption of all four first permanents molars.

The early detection and comprehensive treatment of HPSM remain a priority, since the first permanent molars, as well as the second primary molars, share a period of amelogenesis, and, therefore, the periods of mineralization overlap [10,13,14,15,18]. Furthermore, several authors consider that HSPM could be predictive for MIH [10,13,14,17]. Notwithstanding, MIH notice in children did not present HSPM, thus indicating that the lack of the opacities in primary dentition does not rule out the semblance of MIH [1,9,10,12,13,14,15,16,17,18,20]. In fact, a recent systematic review states that the presence of HSPM is predictive for MIH, although the results must be interpreted with caution [13]. The affected teeth show as chalky white, yellow/creamy, or brown areas of different sizes, and have lower tooth mineral density, leading most likely to post-eruptive breakdown, high rates of carious lesions, and sensitivity, as well as failed restorations due to the poor adhesion of the material, which can lead to the worst scenery of irreversible pulpitis, and therefore to root canal treatment, or even extraction [1,15,16]. The management of HSPM is challenging, especially in non-cooperative and very young children, with the application of nitrous oxide sedation or even general anesthesia sometimes being necessary [6].

The literature shows several treatment options which can be applied in HSPM cases. The application of glass ionomer and build-up using composite resin are very conservative approaches, but this last one has a reduced bonding strength of the composite resin to hypomineralized enamel, which has been unsuccessful in the long term [6]. In severe cases, full coverage with stainless steel or zirconia crown is the best long-term choice [6]. This may be the treatment of choice, because it promotes the preservation of pulp vitality, maintains a proper occlusion, and allows an appropriate eruption of the first permanent molars. The ‘Hall technique’ has been considered a good alternative for the placement of stainless-steel crown [6]. This minimally interventive conservative approach with no tooth preparation was opted for, based on the severity of the damage duo to HSPM and, thereby, the greater need for full-coverage crown to restore the form, function, and longevity of the affected teeth [6]. Moreover, preformed metal crowns can prevent recurrent dental caries, improve oral hygiene, and reduce dentinal hypersensitivity [6].

The second primary molars are of the utmost importance in the occlusion, since they have an eruption guide for the eruption of the first permanent molars [16,21]. Several effects of premature space loss are mentioned, such as dental crowding, ectopic eruption, impaction of the permanent tooth, crossbite, center line discrepancies, and, in cases of premature loss of deciduous second molars, tipping of the first permanent molar can occur [16,22,23,24,25]. Therefore, it is essential to maintain the second primary molar until the first permanent molar tooth and the successor erupt successfully.

## 4. Conclusions

The present case demonstrates that metallic crowns provide a long-term clinical stability in a severe case of hypomineralized primary second molars. Occlusion was successfully maintained and allowed an appropriate eruption of the first permanent molars, avoiding functional and aesthetic negative outcomes. This case stresses the importance of the presence of primary second molars, as well as a predictable and long-term treatment.

## Figures and Tables

**Figure 1 children-08-00923-f001:**
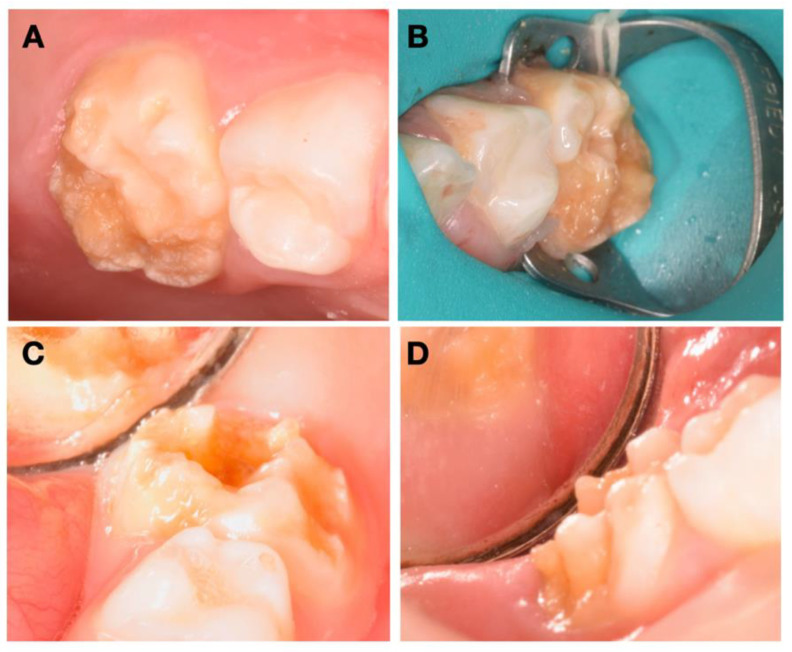
Intraoral pretreatment photographs illustrating preoperative (**A**) occlusal and buccal clinical view of 55, (**B**) occlusal and buccal clinical view of 65, (**C**) occlusal, buccal, and lingual clinical view of 75, and (**D**) buccal clinical view of 85.

**Figure 2 children-08-00923-f002:**
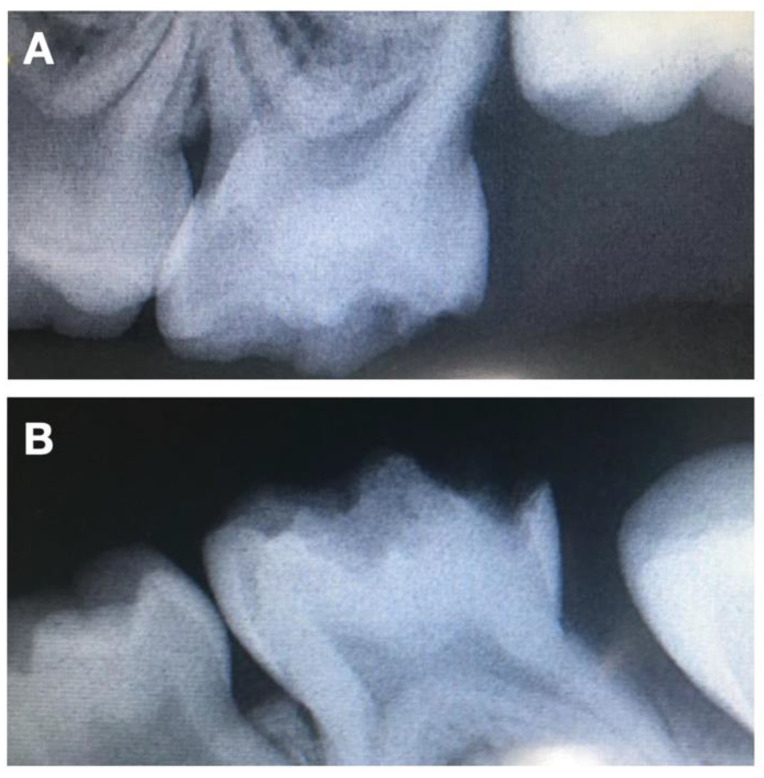
Pretreatment radiographs illustrating preoperative (**A**) periapical X-ray of 65 and (**B**) periapical X-ray of 75.

**Figure 3 children-08-00923-f003:**
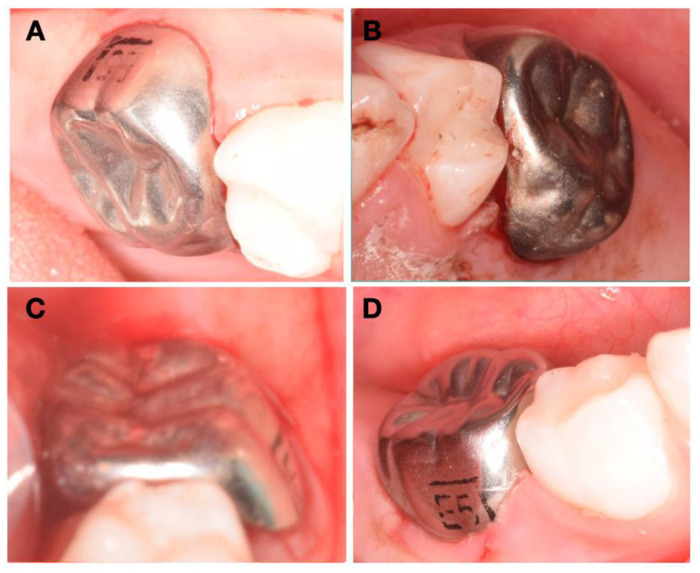
Intraoral post-treatment photographs illustrating preoperative (**A**) occlusal and buccal clinical view of 55, (**B**) occlusal and buccal clinical view of 65, (**C**) occlusal, buccal, and lingual clinical view of 75, and (**D**) buccal clinical view of 85.

**Figure 4 children-08-00923-f004:**
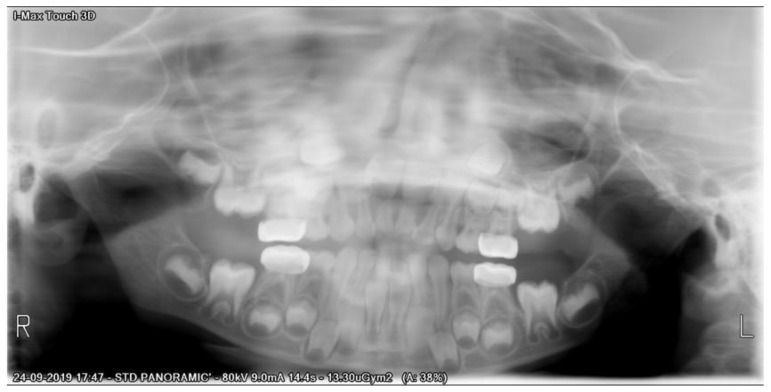
Panoramic X-ray of three-month follow-up.

**Figure 5 children-08-00923-f005:**
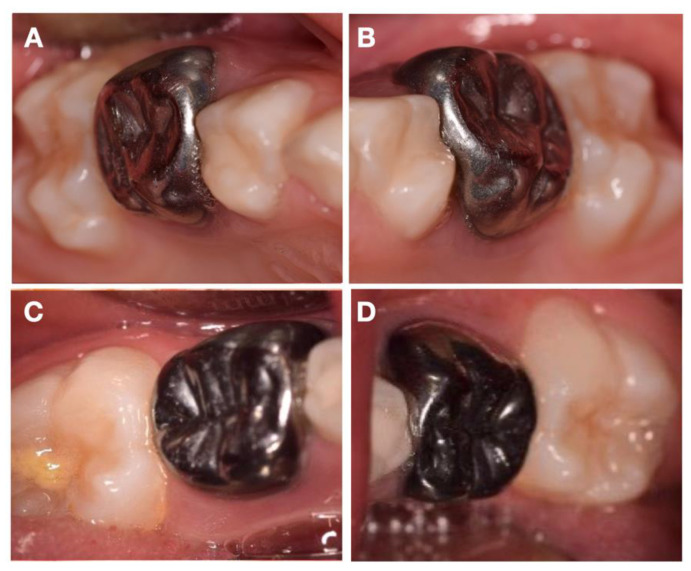
Intraoral post-treatment photographs illustrating four-year follow-up of (**A**) 55, (**B**) 65, (C) 75, and (**D**) 85.

**Figure 6 children-08-00923-f006:**
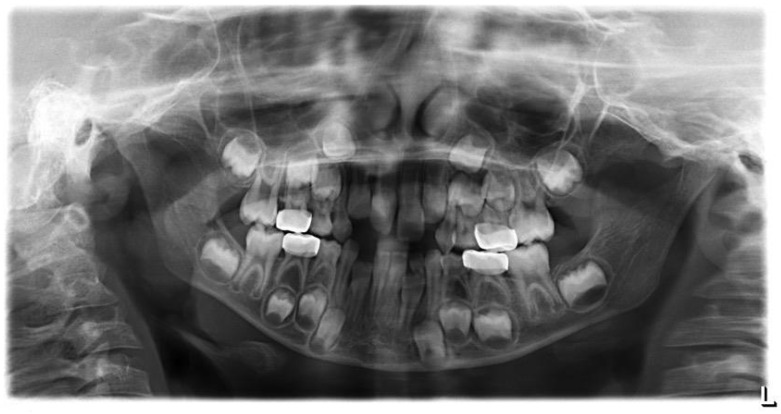
Panoramic X-ray of four-year follow-up.

## Data Availability

Not applicable.

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
