# Peer review of "A Four-Year Follow-Up Case Report of Hypomineralized Primary Second Molars Rehabilitated with Stainless Steel Crowns"

_children, 2021, doi:10.3390/children8100923_

Round 1

Reviewer 1 Report

Introduction: DDS are usual in primary and permanent dentitions... a better expression would be: DDS are frequently observed in primary andd permanent dentitions...

 In  line 29 MIH  is  defined as a systematic hypomineralization - to my knowledge the proper word id systemic.

I suggest to  the authors  to emphasize the need of using SSC (stainless steel crowns) as a  rule in hypoplastic  second primary molars to preserve  pulp vitality, maintain  proper occlusion and allow an appropriate eruption of the first permanent molars.

Author Response

We are delighted to submit our revised manuscript.

We have considered the editorial and reviewers’ comments. Those were considered valuable and useful for revising and improving our paper. We have studied comments carefully and have made corrections accordingly. Modified parts are highlighted in the revised paper. Below is our point-to-point response to each specific comment. Please let us know if anything remains unclear.

We hope the revised manuscript will be in acceptable format for your journal.

Introduction: DDS are usual in primary and permanent dentitions... a better expression would be: DDS are frequently observed in primary and permanent dentitions...

Our answer: We appreciate this remark, and we have revised accordingly.

In  line 29 MIH  is  defined as a systematic hypomineralization - to my knowledge the proper word id systemic.

Our answer: We appreciate this remark, and we have changed to “systemic”.

I suggest to the authors  to emphasize the need of using SSC (stainless steel crowns) as a  rule in hypoplastic  second primary molars to preserve  pulp vitality, maintain  proper occlusion and allow an appropriate eruption of the first permanent molars.

Our answer: Following your important suggestion, we emphasize this by adding: “This may be the treatment of choice because promotes the preservation of pulp vitality, maintains a proper occlusion and allows an appropriate eruption of the first permanent molars.”

Reviewer 2 Report

The manuscript is well written and elaborated in detail but I have few suggestions

  1. Introduction is too short and the concept is scattered. Please elaborate in detail the problem and your purpose of doing this research. Introduction should not be less than one page and not more than one and half page.
  2. The figures added to the manuscript are well represented but I will suggest to improve the quality of the image.
  3. Conclusion is too short. 

Author Response

We are delighted to submit our revised manuscript.

We have considered the editorial and reviewers’ comments. Those were considered valuable and useful for revising and improving our paper. We have studied comments carefully and have made corrections accordingly. Modified parts are highlighted in the revised paper. Below is our point-to-point response to each specific comment. Please let us know if anything remains unclear.

We hope the revised manuscript will be in acceptable format for your journal.

The manuscript is well written and elaborated in detail but I have few suggestions

  1. Introduction is too short and the concept is scattered. Please elaborate in detail the problem and your purpose of doing this research. Introduction should not be less than one page and not more than one and half page.

Our answer: We elaborated more the Introduction section as requested. We hope this meets your expectations.

  1. The figures added to the manuscript are well represented but I will suggest to improve the quality of the image.

Our answer: We appreciate this remark, however, we have further appended images TIFF, and the lower quality may be due to an informatic alteration of the submission platform.

  1. Conclusion is too short. 

Our answer: We have further extended the conclusion section by adding: “This case stresses the importance of the presence of primary second molars, as well as a predictable and long-term treatment.”.
